# Reproductive and Endocrine Outcomes in a Cohort of Danish Women following Auto-Transplantation of Frozen/Thawed Ovarian Tissue from a Single Center

**DOI:** 10.3390/cancers14235873

**Published:** 2022-11-29

**Authors:** Lotte B. Colmorn, Anette T. Pedersen, Elisabeth C. Larsen, Alexandra S. Hansen, Mikkel Rosendahl, Claus Yding Andersen, Stine G. Kristensen, Kirsten T. Macklon

**Affiliations:** 1The Fertility Clinic, Rigshospitalet, University Hospital of Copenhagen, 2100 Copenhagen, Denmark; 2Gynecological Department, Rigshospitalet, University Hospital of Copenhagen, 2100 Copenhagen, Denmark; 3Laboratory for Reproductive Biology, Rigshospitalet, University Hospital of Copenhagen, 2100 Copenhagen, Denmark

**Keywords:** fertility preservation, ovarian tissue transplantation, endocrine function, reproductive outcome

## Abstract

**Simple Summary:**

Ovarian tissue cryopreservation and transplantation is a method used to restore fertility in women with ovarian insufficiency after chemotherapy or irradiation. However, usage of the preserved tissue is low, and more data are needed to identify the strengths and weaknesses of the procedures, allowing for us to improve all steps, from patient selection, surgical and laboratory techniques, protocols for fertility treatment and, finally, live birth rates after ovarian tissue transplantation. This study reports the reproductive outcome and hormonal recovery following ovarian tissue transplantation and evaluates possible predictors of the chance of pregnancy in a cohort of 40 women from a single experienced center in Denmark.

**Abstract:**

Ovarian tissue cryopreservation (OTC) is a method of fertility preservation in girls and young women prior to gonadotoxic treatment. It is a safe and promising method to restore fertility. The initial recovery of endocrine function is high, but the longevity of the grafted tissue varies. In this single-center, combined retro- and prospective cohort study, we report the reproductive outcome and hormonal recovery following ovarian tissue transplantation (OTT) and evaluate possible predictors of the chance of pregnancy. The study includes 40 women from eastern Denmark undergoing 53 OTTs between 2003 and 2021. Permission to obtain retrospective data was given by the Danish Patient Safety Authorities and prospective data-collection by informed consent. Initial recovery of endocrine function was seen in 18/19 women with POI, and ongoing function of the grafted tissue in 7/14 two years from OTT. Live birth rate (LBR) was 41%, with 20 children to 39 women trying to conceive. Women who conceived had higher AFC at the time of OTC than women who did not (*p* ± 0.04). Repeated transplantations were not successful in terms of delivery. Half of all pregnancies were achieved by ART, but PRs were lower after ART than by spontaneous conception. LBRs after OTT are encouraging. Chance of pregnancy after OTT is correlated to ovarian reserve at OTC. Repeated transplantations were not successful in terms of unfulfilled pregnancy wish.

## 1. Introduction

Cryopreservation of ovarian tissue (OTC) has, during the last two decades, gained ground as a method of fertility preservation for girls and women at risk of loss of ovarian function due to iatrogenic causes such as treatment with chemo- or radiation therapy, or excessive ovarian surgery. In many countries, such as Denmark, Belgium, and the US, with solid fertility preservation programs, it has moved from an experimental to a standard procedure [1,2]. However, it is used for selected patient groups only and vitrification of oocytes and embryos remains the standard procedure for fertility preservation. OTC is performed by surgically removing healthy ovarian tissue before a potentially sterilizing treatment, cryopreserving, and later transplanting it back to the same individual. It is possible to restore the ovarian function, not only by enabling the woman to become pregnant but also by restoring her endogenous hormone production.

At present, it is unknown how many girls and women have had OTC worldwide, and how many have returned for OTT. From large fertility preservation cohorts, the return rates are reported to be between 3.4% and 19% depending on follow-up time [3,4,5]. The efficacy of OTT varies, but recent reviews and meta-analyses from large centers and systematic literature searches have shown live birth rates ranging from 26 to 38% [2,6,7]. Less is known about reproductive endocrine function after OTT, the duration of the ovarian grafts, and results after in vitro fertilization (IVF) in women with OTT. In a recent meta-analysis looking at pooled data from several studies, Khattak and colleagues found an increase in oestrogen and a decrease in Follicle Stimulating Hormone (FSH) and Luteinizing Hormone (LH) post-transplant compared to pre-transplant levels, indicating a return of endogenous hormone production after OTT [7]. Anti-Müllerian Hormone (AMH) concentrations, on the other hand, often remain low after OTT [8] and do not seem to show any correlation with the ability to obtain a pregnancy [7]. Most pregnancies after OTT are achieved spontaneously [2,7,9], but it seems that for those women/couples in need of ART, for known or unknown reasons, live birth rates after OTT are reduced compared to those without infertility issues. Studies on IVF in women with OTT have found live birth rates (LBR) are as low as 3.9% per cycle, which is considerably lower than in the general IVF population. This is often attributed to poor response to stimulation, a high empty follicle rate and a high miscarriage rate [10,11].

The longevity of the grafts can vary from a few months to more than 10 years, although there seems to be an average duration of between 2.5 to 5 years [2,7].

Although we over time gain more and more experience and knowledge about OTC and OTT from an increasing number of publications, there is still room for improvement. More data needs to be collected and analyzed, to identify the strengths and weaknesses of the procedures, and hereby allow us to improve all steps from patient selection, techniques of cryopreservation and thawing, and ART protocols, to live birth rate after OTT.

The purpose of this study is to present a descriptive analysis of a cohort of women going through OTT from a single, specialized center in Denmark, with 23 years of experience in fertility preservation. Our center is one of three centers performing OTC in Denmark. Nationally, a total of 1186 girls and women had OTC from 1999 to 2020, from where 117 women underwent OTT [12].

## 2. Materials and Methods

In January 2020, the ‘Copenhagen fertility preservation cohort research study’ was initialized at the Fertility Department at Rigshospitalet, Copenhagen, Denmark. Women seeking fertility preservation at our clinic up until Jan 2020 were retrospectively included in the cohort, while women from January 2020 and onwards were included prospectively. Indications for fertility preservation, as well as freezing and thawing procedures for ovarian tissue, have previously been described [12]. Information on the indication and method for fertility preservation, oncologic history, gonadotoxic treatment, gynecologic and obstetric history, comorbidities, and endocrine parameters (FSH (IU/L), LH (IU/L), oestrogen (nmol/L), and progesterone (nmol/L)) and ovarian reserve markers (antral follicle count (AFC), anti-müllerian hormone (AMH (pmol/L)) before and after OTC and OTT was collected from paper-based and electronic medical records from public Danish hospitals and entered in a RedCap research database hosted at Rigshospitalet, in Copenhagen Denmark. Data on pregnancies and deliveries were obtained from medical records.

In our cohort, all women returned for OTT due to an unfulfilled pregnancy wish. For the present analysis, the ovarian function before and after OTT was categorized into three subgroups: premature ovarian insufficiency (POI), unstable ovarian function (UOF) and ongoing/restored ovarian function with low ovarian reserve (LOR). POI was defined as hypogonadism, increased FSH levels above 25 IU/L and amenorrhea > 4 month according to the ESHRE criteria [13]. UOF was defined as fluctuating FSH levels around 25 IU/L and continuous regular or irregular cycles. LOR was women with an unfulfilled pregnancy wish despite ongoing/restored ovarian function with normal FSH levels and ongoing menstrual cycles, all with AMH values below 5 pmol/L.

The center at Rigshospitalet cryopreserved ovarian tissue in 441 women from 1999 to March 2021. From 2004 to March 2021, 40 women underwent 53 transplantations of ovarian tissue at our center. All women had an entire ovary removed laparoscopically for cryopreservation. In case of concern regarding risk of malignant recurrence or serious complications during surgery or pregnancy, the oncologist or haematologist responsible for the patient was consulted. OTT was not offered in case of concerns. All but one OTT procedure was performed as a combined laparoscopy/mini-laparotomy. Tissue was transplanted to the remaining ovary in all cases, and to alternative sites when deemed safe and possible (ovary and peritoneal pocket, *n* = 29; ovary only, *n* = 11). None of the women experienced major complications from the transplantation. After OTT, the women were seen on a regular basis, with four-week intervals, until signs of ovarian activity were documented by either the presence of antral follicles on ultrasound or by improvements in endocrine markers. Women with a wish for pregnancy and no recognized infertility issues were encouraged to try for a spontaneous pregnancy from six to twelve months following OTT. Women with perceived infertility factors (e.g., tubal factor or poor semen quality), as well as single women and women in same-sex relationships, went straight to ART. Most of the included ARTs were performed at our own center at Rigshospitalet, and only a very few treatments were performed in private fertility clinics. For the present study, the end of follow-up (EOF) was December 2021.

The reproductive outcomes from before December 2017 have previously been published by Gellert et al., Jensen et al., and Dolmans et al., as part of the data from the national cohort, including data on 25 women with 13 pregnancies and 7 deliveries from our center [2,5,9]. We now expand the results from our clinic up to December 2021.

### Ethics

Approval for retrospective data collection was obtained from the local data management committee (jrn 3-3013-2790/1). Prospective data collection was further obtained by informed written consent from the patients. 

## 3. Results

During the study period, 40 women underwent 53 OTTs. Basic characteristics are presented in Table 1. Indications for OTC were malignant (85%) or benign diseases (15%). The most common malignancy was breast cancer, followed by hematological malignancies. Other malignancies included rectal cancer, Ewing’s sarcoma, and thymus cancer. Benign indications included multiple sclerosis, aplastic anemia, and paroxysmal nocturnal hematuria (Table 1). Two women had chemotherapy before OTC: one with non-Hodgkin’s lymphoma who had her first series of Cyclophosphamide, Doxorubicin, Vincristine, and Prednisolone (CHOEP) and one with Hodgkin’s Disease who had six series of Doxyrubicine, Bleomycine, Vinblastine, and Procarbazine (ABVD).

The mean age at the time of OTC was 28.9 (SD 5.5, range 17–37). The patient characteristics at the time of OTC are shown in Table 1. The mean time from OTC to the first OTT was 4.2 years (SD, 1.3–15.4 y), and the mean age at first OTT was 33.7 (SD 5.4, 22–44). At the first OTT, 19 women had POI, 10 women had unstable ovarian function and 11 had ongoing ovarian function (Table 1). Of the 40 women, 12 (30%) had a second and one (2.5%) a third OTT during follow-up.

The mean follow-up time from the first OTT to the end of study follow-up (31 December 2021) was 6.1 ± 4.7 years (from 11 months to 18.7 years). However, some women were lost to follow-up before EOF, and the mean follow-up time from the first OTT to the last relevant note in the medical record was 4.1 ± 2.8 years (from 4 months to 11.4 years), and from the last OTT 2.8 years ± 1.6 (11 months to 6.7 years).

### 3.1. Pregnancy and Live Birth Rates following OTT

One woman became single after OTT and no longer pursued pregnancy. Of the remaining 39 women with a pregnancy wish, 22 were pregnant, with 29 pregnancies, and 11 gave birth to a total of 15 children; four women delivered twice. In addition, five women had an ongoing pregnancy in the last trimester at the EOF, which all resulted in live births at the time of writing. The overall PRs and LBRs (incl ongoing) per woman were 56% and 41%, respectively (Table 2). The overall pregnancy loss rate was 31%, incl. 6 biochemical pregnancies, 1 early miscarriage, and 2 ectopic pregnancies. A total of 48% of all pregnancies and 40% of all deliveries were achieved following ART (Table 3). The pregnancy loss rate (incl. ectopic pregnancies) was 43% following ART, and 20% following spontaneous conception (Table 3). The mean time to delivery from the first OTT was 29 ± 23 months, with a variation between 11 and 103 months (8.5 years). Twenty-six pregnancies and 19 deliveries were reported after the first OTT (incl. ongoing pregnancies), and 3 pregnancies and one delivery after the second OTT. Four women had 2 children, of whom one had her second child following the second OTT. Repeated transplantations were not successful in terms of delivery, since no woman without a live birth after the first transplantation succeeded with a live birth after the second transplantation. Of the three women who became pregnant after the second OTT, one woman became pregnant after both the first and second OTT, but the first pregnancy was ectopic and the other was a biochemical pregnancy. One woman achieved her first pregnancy after the second OTT, resulting in an early miscarriage, and the last woman delivered after both the first and second OTT.

Women who conceived after OTT were compared to those who did not. The mean AFC was significantly higher (*p* = 0.04) in women with than without pregnancy (Table 1). However, there was no difference in mean age and AMH at OTC, indication for OTC, the number of women with pelvic radiation or POI status after gonadotoxic treatment between pregnant and non-pregnant women (Table 3). A total of four women had pelvic radiation before OTT. However, only one had high doses of about 50 GY due to rectal cancer, while the other three had low-dose TBI in combination with bone marrow transplantation for benign or malignant disease. Two of the women with TBI achieved a pregnancy, while the woman with rectal cancer did not become pregnant.

### 3.2. Fertility Treatment

During follow-up, 31 women underwent a total of 194 treatments following OTT (mean 6.2 ART cycles per woman), including 18 treatments with donated oocytes in seven women. 

Excluding treatments with oocyte donation, 30 women had a total of 176 treatments: 121 IVF/ICSI cycles (69%), 49 inseminations with husband’s or donor sperm (28%) and six frozen embryo transfers (FET) (4%). No pregnancies were observed following insemination.

Of the 121 IVF/ICSI and 6 FET cycles, 77 were canceled (76 IVF/ICSI and 1 FET) giving a cancellation rate of 61% and embryo transfer was possible in 50 cycles (Table 2). In 43 cases (56%), no transfer was possible due to arrested embryos or atretic oocytes; in 19 cases (25%), the cycles were canceled before egg retrieval due to the lack of growing follicles (*n* = 12) or unfavorable conditions before or during stimulation (high FSH at start of stimulation, irregular menstrual bleedings, premature ovulation); in 12 cases (15%), the reason for the cancellation was unknown; in 3 cases, the follicles were empty (4%). A total of 14 pregnancies were registered in 11 women, with a total of 8 deliveries in 6 women. The overall clinical pregnancy rate was 7.8% with an LBR of 4.9% per started cycle and a clinical pregnancy rate of 20.4% with an LBR of 16.4% per transfer (Table 4). Overall, about one in five women in need of IVF achieved a live birth. Two women conceived naturally after IVF treatment.

### 3.3. Endocrine Function of the Grafted Tissue in Women with POI

Of the 40 women who underwent a first OTT, 19 had a diagnosis of POI. Of these, all but one achieved an initial decrease in FSH levels to values ≤ 25 IU/L and resumption of menstrual cycle after OTT (18/19, 95%). The mean ± SD time to normalization of the FSH levels was 5.2 ± 1.0 months (min 3.5, max 7.0). The woman with total graft failure had breast cancer at the age of 32 years. At the time of OTC, she had a low AMH 3.4 pmol/L, AFC 3, and no use of hormonal contraception. She had one previous pregnancy and delivery before her cancer diagnosis. 

The follow-up time on endocrine data after the first OTT varied, with a mean ± SD duration of 4.9 ± 2.9 years, with a range from one to eleven years. Figure 1 shows the number of women with and without normal endocrine function in months after the first OTT.

One year from the first OTT four women had recurrent POI including the women with total graft failure.

Two years from the first OTT, a total of seven women had POI, while seven had ongoing endocrine function of the grafted tissue. One woman had a second transplantation within two years due to IVF failure. Four women had less than two years of hormonal follow-up, where three had an ongoing pregnancy and one had normal hormonal levels at the EOF 1.6 years from the first OTT (Figure 1).

Three years after the first OTT, another woman had POI. Three women had normal endocrine function, and a further two were lost to follow-up: one with an ongoing pregnancy and one with normal hormonal levels at end of follow-up.

Four years from the first OTT, no further women had POI, one woman had an ongoing pregnancy, and two had continuously normal endocrine function of the grafted tissue.

After five years, one more woman had POI, leaving a total of 9 women with POI. The last woman with complete hormonal data presented with an ongoing ovarian-graft function until seven years after the first OTT, when she decided to have a second OTT to boost her ovarian function for a second child.

### 3.4. Repeated Transplantations

Nine of the 19 women with initial POI at the first OTT underwent a second, and one a third transplantation with a mean ± SD of 3.3 ± 1.9 years between (min 1, max 7). The indication of a second or third procedure was recurrent POI in three women, insufficient ovarian function with perimenopausal FSH levels in three women, and IVF failure in three women. FSH levels in the nine women from the first OTT to EOF are shown in Figure 2.

Of the six women with a second transplantation due to POI or insufficient ovarian function with perimenopausal FSH levels, two had the second procedure within two years, one within three years, and three about 4.5 years following the first OTT (Figure 2). Of these, one woman experienced total graft failure (Figure 2C), one had poor recovery of ovarian function with POI about 8 months after the second transplantation (Figure 2F) and one had recurrent ovarian insufficiency with perimenopausal FSH levels from about 2 years after the second transplantation (Figure 2B). The remaining six women had normal FSH levels during follow-up, with a follow-up period that varied from 5 to 50 months (mean ± SD: 23.5 ± 22.1). Of these, one woman had an oophorectomy about one year after the second transplantation due to the national recommendations for women with BRCA mutation and one woman had a third transplantation 11.5 years following her second OTT due to continuous IVF failure. No hormonal data were available following the third procedure.

The longest registered period with ongoing endocrine function after the first OTT was 7.2 years; when the woman had her second OTT, this woman also presented with the longest registered total period with ongoing endocrine function after two procedures at the end of follow-up, 7.6 years after her first OTT.

### 3.5. Relapse of Malignancy after OTT

Five women were registered with recurrent malignant disease following first-line chemotherapy, two following OTT. One woman with thymus cancer with a high recurrence risk had three relapses within 6 years from OTC. One was following first-line chemotherapy, and one following first and second OTT, respectively. Another woman with breast cancer had local relapse in her breast 6 years after OTT. She had a total oophorectomy three years after relapse. None of the relapses were found to be related to the grafted tissue. No deaths were registered in the population.

## 4. Discussion

This cohort study reports the reproductive outcomes in 40 women and the endocrine outcomes in 19 women undergoing OTC and OTT with the purpose of preserving fertility. Twenty-two women achieved a total of 29 pregnancies, and 16 gave birth to one or more healthy baby. Pregnancy and livebirth rates were higher after natural conception than ART (60 vs. 40%). The pregnancy rates in our cohort are among the highest in the literature, where PRs vary between 33 and 50% and LBRs vary between 23 and 42% [2,6,7,14,15]. The variation in success rates among centers could be due to differences in population characteristics such as age, indications, or pelvic radiation after OTC. Studies have shown limited success in pregnancy and live births in women older than 35 years at the time of OTC, and women who become pregnant are, in general, younger than women who do not [2,7,16,17,18]. Our cohort primarily consisted of women less than 35 years of age, and we did not find any difference in mean age at OTC between pregnant and non-pregnant women. Age is related to ovarian reserve [19], but individual variations are present, which is why ovarian reserve markers at the time of OTC could possibly help us to predict the efficacy of OTT in terms of pregnancy and the longevity of the grafted tissue. It is known that measurement of AMH after OTT is of limited value to predict pregnancy and the longevity of the tissue [8,20,21]. However, to the best of our knowledge, no studies have evaluated the relationship between ovarian reserve markers (AFC and AMH) at the time of ovarian tissue cryopreservation and the chance of pregnancy after OTT. In our study, we found that pregnant women had a significantly higher mean AFC at OTC than non-pregnant women, which supports the theory that women with a higher ovarian reserve prior to ovarian cryopreservation will also have more viable follicle after OTT and an increased chance of conceiving. Our results did not show any difference between pregnant and non-pregnant women in relation to mean AMH at OTC. However, AMH prior to OTC was only available in about half of the women, and the result could be due to the lack of power. Pelvic radiation is another factor linked to an impaired chance of pregnancy and delivery due to uterine damage with fibrosis and poor vascularization, leading to an increased risk of miscarriage, placental dysfunction, preterm birth and low birth weight after doses as low as 2.5 GY [22,23,24,25,26]. In a systematic review, the et al. advised against pregnancy from doses above 25 GY in childhood and 45 GY in adults [25]. In our study, only one woman was exposed to high radiation doses to the uterus (>45 GY), while the other three women had lower doses of TBI, and two of these achieved pregnancy. This is consistent with data in a review from five European centers (FertiPROTEKT, Denmark, Spain, France and Belgium), where only women with reduced radiation doses below 45 GY gave birth [2].

Repeated transplantation is considered in cases of graft malfunction or IVF failure with unfulfilled pregnancy wish. In our study, 12 women had repeated transplantation, out of which three conceived. However, only one woman had a successful pregnancy and live birth after the second transplantation. This is in line with reports from other studies, in which live birth rates after repeated transplantations are limited, and it is our opinion that women should be informed about the poor success rate before opting for a second transplantation [14,18].

The recovery of endocrine function with a decrease in FSH levels below 25 IU/L was achieved in nearly all women following OTT, which is in accordance with results from other experienced centers [2,7,27]. Long-term follow-up on the longevity of the endocrine function is poorly reported in the literature and, in many cases, limited to casuistic reports. In our study, we were not able to determine the full potential of repeated procedures since the follow-up on hormonal data was very heterogenous. Our study suggests that almost 50% of women require repeated transplantations to maintain steady endocrine function for more than 2–3 years. We saw that the second transplantation in women with recurrent POI after the first OTT increased the period with normal endocrine function by more than one year in 3/6 women. In a study by Dolmans et al., they reported a 5-year graft survival rate of 55% after one or more transplantations in a subgroup of 45 women with full hormonal data after five years [2]. However, they did not report how many of the women had more than one procedure during the five-year follow-up. The potential of OTT as an alternative to conventional HRT is not yet fully explored, and further studies with detailed long-time hormonal follow-up are needed before OTT can be recommended as hormonal replacement therapy.

In our cohort, only 1 in 5 women in need of IVF after OTT achieved a live birth. This is in line with reports of the reproductive outcome after OTT with IVF in the literature, with high cancellation rates (13–39%), high empty follicle rates (25–35%) and high pregnancy loss rates (37–60%) [2,10,11,28]. The reported pregnancy rates vary from 3.9 to 19.3% and live birth rates of 3.9–14% per started cycle, and 7.4–50% and 5–37.5% per embryo transfer, respectively [10,11,28]. The reason for the poor reproductive outcome in women needing IVF is not fully understood but could be partly explained by the inherently low ovarian reserve in women after OTT, which is a factor that is known to be linked to poor IVF outcome in general. A suboptimal luteal phase with inadequate progesterone levels could also be a partial explanation. This emphasizes the need for further studies, to improve the chance of pregnancy and delivery after OTT with IVF.

Autotransplantation of ovarian tissue to women with previous malignant disease includes a potential risk of reintroducing malignant cells to the woman, with a theoretical risk of relapse of the original disease. However, clinical data have shown that only 9 of 230 women (4%) with malignant disease relapsed after OTT, where the relapse was found to not be directly related to the transplanted ovarian tissue in eight of these cases [5]. The risk of transmission of malignant cells is considered to be low for breast cancer (stage I–III), Hodgkin’s lymphoma, osteosarcoma and non-genital rhabdomyosarcoma, moderate for breast cancer (stage IV), gastrointestinal cancer, endometrial cancer, cervical adenocarcinoma, non-Hodgkin’s lymphoma and Ewing’s sarcoma, and high for leukemia, neuroblastoma, Burkitt’s lymphoma and ovarian tumors [29,30]. In our cohort, two women (5%) had a relapse of malignant disease during follow-up, but neither of these cases appeared to be related to the transplanted tissue.

Although OTC with OTT shows promising results for fertility preservation and restoration, the return rates for utilization of the ovarian tissue remain low, between 4 and 9% worldwide [3,4,12,31,32]. The reason for this remains to be clarified. One explanation could be that the girls/women are still too young to desire parenthood, as suggested in the paper from Kristensen et al., where the return rates for a subgroup in the expected fertile age were 13%, compared to 9% for the entire population [12]. Other reasons could be recommendations against pregnancy due to general health or women’s concerns about parenthood if they have a history of cancer and risk of recurrent disease. However, the most likely reason is probably that fewer than expected become menopausal following gonadotoxic treatment, demonstrating the need for better patient selection. More studies are needed to optimize patient selection, and to clarify if a greater portion of women than have ongoing ovarian function following gonadotoxic treatment than expected. In that case, it is important to consider the potential harm of the OTC procedure to future endocrine and fertility outcomes when one ovary is removed, halving the ovarian reserve, when standard fertility preservation options with cryopreservation of eggs/embryos are available and were shown to be equal in terms of pregnancy and livebirth rates [17].

The strength of our study is the single-center set-up, with uniform standards for all patients. The team of doctors and laboratory staff handling the tissue was small, increasing the experience and skills of the involved health care professionals. We believe that it is important to the patients that they are seen and cared for by the same doctors throughout the whole process. In Denmark, fertility preservation in relation to gonadotoxic treatment is offered in specialized centers at public hospitals free of charge. The patients at our center are offered a follow-up on ovarian activity and fertility counselling at the end of chemotherapy, with the possibility of IVF treatment at our clinic if needed. This means that the patients are managed all the way from initial counselling to OTC and OTT and during follow-up, with information on subsequent IVF and pregnancies [33,34]. Our database builds on data from medical records from public hospitals and contains all available data, which minimizes the risk of missing data. Nevertheless, although there were twenty years of follow-up from the first OTC, overall return rates are still low [12], and the retrospective design results in missing values and heterogeneous follow-up compared to a prospectively designed follow-up study.

## 5. Conclusions

Our results support the use of OTT for fertility restoration in women with previous gonadotoxic treatment with a high initial recovery rate of the endocrine function and a LBR of 41% per woman. However, compared to natural conception, pregnancy, and live birth rates after IVF in this group of patients are poor and leave room for improvement. Repeated transplantations prolong the endocrine function, but seem less successful in terms of pregnancies, both spontaneously and after IVF. More data on the long-term duration and endocrine stability of the grafted tissue is needed before recommendations can be given regarding the use of OTT for hormonal replacement therapy.

## Figures and Tables

**Figure 1 cancers-14-05873-f001:**
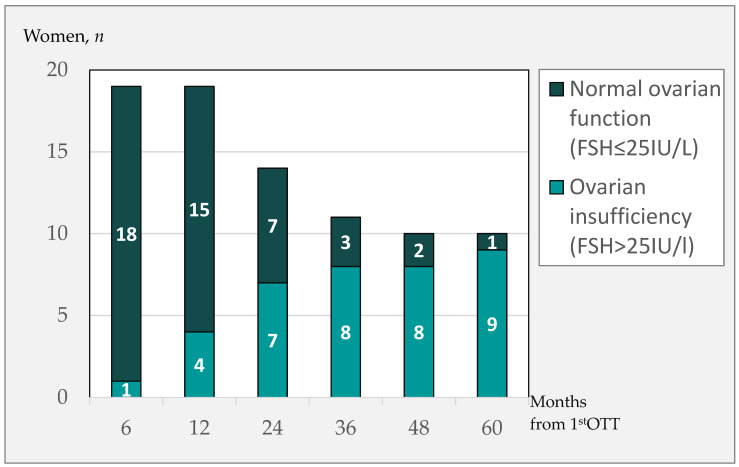
Endocrine function in women following the first OTT.

**Figure 2 cancers-14-05873-f002:**
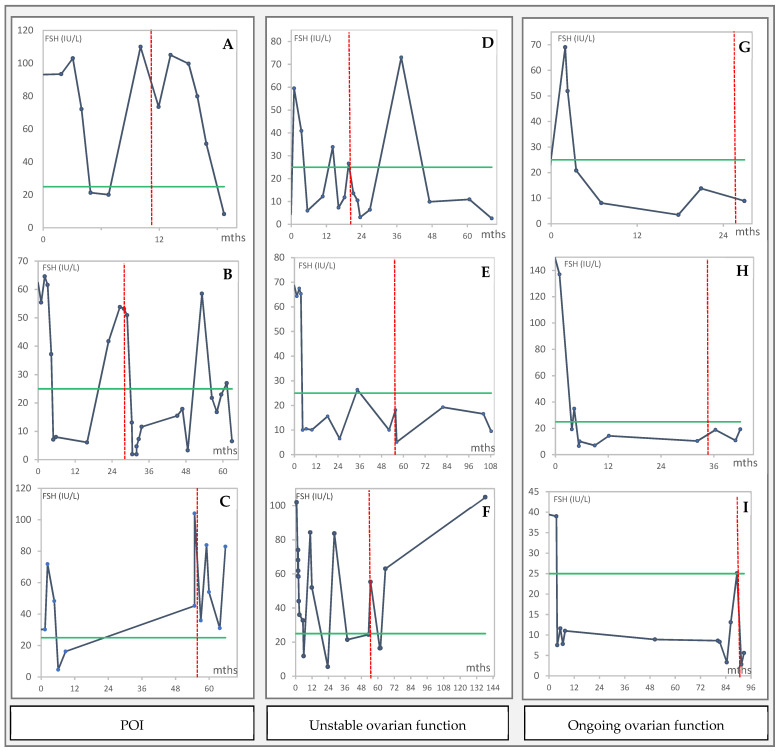
Fluctuation of FSH (IU/L) in months from the first OTT in the nine women with a second OTT, who had POI at the first OTT. The women (**A**–**I**) are grouped by indication for the second OTT ((**A**–**C**): recurrent POI, (**D**–**F**): unstable ovarian function, (**G**–**I**): ongoing ovarian function). The red vertical dotted line indicates the time for the second OTT. The green horizontal line indicates the reference value for FSH = 25 IU/L.

**Table 1 cancers-14-05873-t001:** Characteristics of women going through ovarian tissue transplantation, overall and with/without pregnancy.

	Women, *n* = 40 (%)	Pregnant, *n* = 22 (%)	Non-Pregnant, *n* = 17 (%)	*p*-Value
Age at OTC ^a^, mean (SD)	28.9 (5.5)	28.9 (5.0)	28.7 (6.2)	0.94
15–19	3 (7.5)	2 (9)	1 (6)	
20–24	6 (15.0)	2 (9)	4 (24)	
25–29	9 (22.5)	5 (23)	4 (24)	
30–34	15 (37.5)	11 (50)	4 (24)	
35–39	7 (17.5)	2 (9)	4 (24)	
AMH at OTC ^a^, mean (SD) (Median, IQR)	19.7 (19.9) (15.0, 14.8)	25.8 (24.8) (19.0, 10.0)	10.3 (6.4) (7.2, 7.1)	0.14
AFC at OTC ^a^, mean (SD) (Median, IQR)	15.8 (14.2) (12.0, 10.0)	20.0 (16.7) (16.0, 18.0)	8.7 (4.7) (10.0, 8.0)	0.04
Diagnosis at OTC ^a^				
Breast cancer	19 (48)	11 (50)	8 (47)	
Leukemia (CML)	1 (3)	0 (0)	1 (6)	
Hodgkins disease	7 (18)	4 (18)	3 (18)	
Non-Hodgkins lymphoma	4 (10)	2 (9)	2 (12)	
Other malignancy	3 (8)	1 (5)	2 (12)	
Benign disease	6 (15)	4 (18)	1 (6)	
Previous deliveries	6 (15)	4 (18)	2 (12)	
Pelvic radiation				1.0
Yes	4 (10)	2 (9)	2 (12)	
No	36 (90)	20 (91)	15 (88)	
Age at first OTT	33.7 (5.5)	28.9 (5.0)	28.4 (6.2)	0.76
Time to first OTT (years)	4.2 (2.9)	3.6 (1.2)	5.3 (4.0)	
Ovarian function before first OTT				0.12
POI	19 (48)	9 (41)	9 (53)	
Unstable ovarian function	10 (25)	4 (18)	6 (35)	
Ongoing ovarian function	11 (28)	9 (41)	2 (12)	
Number of OTTs				
1	40 (100)	16 (73)	11 (65)	
2	12 (30)	5 (23)	6 (35)	
3	1 (3)	1 (5)	0 (0)	
ART after OTT (from 39 women)				
Yes	30 (77)	11 (50)	17 (100)	
No ^b^	9 (23)	11 (50)	0 (0)	

^a^ At the time of ovarian tissue cryopreservation. ^b^ Two women conceived naturally during ART, why the total number of women with pregnancy after ART differs from the total number of women without ART.

**Table 2 cancers-14-05873-t002:** Pregnancy and live birth rates in 39 women with a pregnancy wish.

	Women with Pregnancy Wish	Pregnancies (*n* = 29)		Deliveries (*n* = 20)	
	*n*	Women, *n*	Rate % (95% CI)	Women, *n*	Rate % (95% CI)
Overall	39	22	56% (39.6, 72.2)	16	41% (25.6, 57.9)

**Table 3 cancers-14-05873-t003:** Mode of conception and pregnancy outcomes in women after OTT.

Method of Conception	Women with a Pregnancy Wish *n*	Women who Conceived *n* (%)	Women who Gave Birth *n* (%)	Pregnancies *n* (%)	Deliveries *n* (%)	Pregnancy Loss *n* (%)
ART	30	11 (50)	6 (38)	14 (48)	8 (40)	6 (43)
Natural	11	11 (50)	10 (63)	15 (52)	12 (60)	3 (20)

**Table 4 cancers-14-05873-t004:** ART treatments with and without pregnancy and pregnancy outcomes.

	Cycles, *n*			Pregnancy Outcome, *n*					Pr Transfer (%)	
	ALL	Cancelled	Transfer	Ectopic	Early Miscarriage	Bio-Chemical	Deliveries	TOTAL	Clinical PR	LBR
IVF/ICSI	121	77	44	1	1	3	6	11	18.2%	13.6%
FET	6	1	5	0	0	1	2	3	40.0%	40.0%
**TOTAL**	**127**	**78**	**49**	**1**	**1**	**4**	**8**	**14**	**20.4%**	**16.4%**

## Data Availability

The data presented in this study are only available on request from the corresponding author in case of international collaborations and requires approval from the Danish patient safety authorities.

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
