# Peer review of "Reproductive and Endocrine Outcomes in a Cohort of Danish Women following Auto-Transplantation of Frozen/Thawed Ovarian Tissue from a Single Center"

_cancers, 2022, doi:10.3390/cancers14235873_

Round 1

Reviewer 1 Report

The aim of this paper is to report the authors results of ovarian tissue transplantation in 40 women in a single-center in Danmark, regarding reproductive outcome and hormonal recovery and to evaluate possible predictors of the chance of pregnancy.

Several such reports have already been published together with pooled results of the biggest OCT-OTT centers and also metaanalyses. 

The paper has several strengths: 

It is nicely written, tables and figures are clear and complete allowing a clear view of the results. 

A second strenght is the settings of a single center study showing that results might be center related as life birth rates are higher in this group than in pooled centers papers or meta-analyses. 

Also this paper is one of the few giving data on AFC and AMH before OTC and distinguishing results un case of POI, unstable ovarian function and ongoing ovarian function. 

The review is rather complete with analyses of the relevant issues as found in other papers (influence of age, AFC, AMH, previous chemotherapy, radiotherapy, transplantation site, time to recovery, miscarriage rates, comparison of spontaneous pregnancies and pregnancies after ART, effects of second and third grafts, evolution of hormonal levels, potentiel differences between IOP and persistant ovarian function, relapse of malignancy). Conclusions are similar to related papers. 

References are up to date and self-citations are not excessive. 

My hesitation to accept this paper is related to previous publications of the same authors and others. 

A part of the results were already published in 2015 (Jensen et al), 2018 (Gellert et al.) and 2020 (Dueholm et al).

The results of this team are also incorporated in recent reviews (Dolmans 2021, Khattak 2022).

Altogether, despite the fact that it does not give a lot of new scientific data, I think it should be considered for publication because of its strenghts. 

Specific comments: 

It might help understand the differences between figures in different papers of the authors to describe the number of centers performing OTC and OTT in Danmark (ex. Dolmans 2022: 62 OTT from Danmark, Kristensen 2021: 117 OTT in Danmark)

line 29: 18/19 is not very clear as first it says 40 women, it might be better to add 18/19 women with POI

line 30: 8/16: shouldn’t it be 7/14 (cfr figure 1)

line 30: 41%, with 20 children to 16 of 39 women trying to conceive

line 32: Half of pregnancies were achieved by ART, but PR were lower after ART than by spontaneous conception. 

line 309: ‘,who’ should be removed

line 329: (moderate…) no verb in sentence

Author Response

Response to Reviewer 1 comments:

Thank you for the review of our paper! Please find our responses below, marked with red:

  1. My hesitation to accept this paper is related to previous publications of the same authors and others. 

A part of the results were already published in 2015 (Jensen et al), 2018 (Gellert et al.) and 2020 (Dueholm et al).

The results of this team are also incorporated in recent reviews (Dolmans 2021, Khattak 2022).

Response 1. We have now further specified previously published data from our center in the text. The paper by Dueholm et al. does not include data from our center: 

‘The reproductive outcomes from before December 2017 have previously been published by Gellert et al., Jensen et al., and Dolmans et al as part of data from the national cohort, including data on 25 women with 13 pregnancies and 7 deliveries from our center [2,5,9]. We now expand the results from our clinic up to January 2020.'

Specific comments: 

  1. It might help understand the differences between figures in different papers of the authors to describe the number of centers performing OTC and OTT in Danmark (ex. Dolmans 2022: 62 OTT from Danmark, Kristensen 2021: 117 OTT in Danmark)

Response 2. I have for clarification specified the total national numbers of OTC and OTT until 2020 in the manuscript.  

  1. line 29: 18/19 is not very clear as first it says 40 women, it might be better to add 18/19 women with POI

line 30: 8/16: shouldn’t it be 7/14 (cfr figure 1)

line 30: 41%, with 20 children to 16 of 39 women trying to conceive

line 32: Half of pregnancies were achieved by ART, but PR were lower after ART than by spontaneous conception. 

line 309: ‘,who’ should be removed

line 329: (moderate…) no verb in sentence

Response 3. Thank you for the linguistic corrections. They are applied to the manuscript. 

Reviewer 2 Report

Reproductive and endocrine outcomes in a cohort of Danish women following auto-transplantation of frozen/thawed ovarian tissue from a single center. Lotte B. Colmorn1*, Anette T. Pedersen2 , Elisabeth C. Larsen1 , Alexandra S. Hansen1 , Mikkel Rosendahl2 , Claus Yding 5 Andersen3 , Stine G. Kristensen3 , Kirsten T. Macklon1

This paper reports on the outcomes of women who had ovarian tissue cryopreservation (a whole ovary removed) prior to gonadotoxic treatment for predominantly malignant diseases (85% vs 15% for benign diseases), and returned for ovarian tissue transplantation for unfulfilled fertility desires.  This is an update on a subset of the cohort first published by this group in 2012 and updated in 2015.

I think the paper will be of interest to the journal’s target audience, which I assume are HCPs working in the field of cancer.

The paper is well-written, easy to read and with good quality informative graphics.

Given the assumed target audience, I feel care must be taken to ensure the correct context for the presented data.  The paper starts with a strong statement that OTC is an established treatment (lines 20 and 41), and indeed the ASRM has stated this (ref 2 quotes ref 1). However, it is described as innovative treatment by ESHRE (Hum Reprod Open 2020 Nov 14;2020(4):hoaa052. doi: 10.1093/hropen/hoaa052.eCollection 2020.). In the UK, for example, funding is not routinely available for OTC (whereas it is for cryopreservation of eggs/embryos). NICE did not update their recommendation for cryopreservation to include OTC in their most recent update, although this was in 2017. Given that this paper is aimed at non-reproductive specialists, it is important that they are not led to believe that this is standard treatment for fertility preservation in most countries. Cryopreservation of eggs or embryos remains the standard (ASRM, ESHRE, NICE). The authors have chosen not to use such a strong statement in the simple summary.

The audience are likely to benefit from a brief discussion of the pros and cons of OTC vs standard fertility preservation options (cryopreservation of eggs/embryos) for context. It is important to consider the potential harm of the OTC procedure to future endocrine and fertility outcomes (particularly since 1 ovary was removed, halving the ovarian reserve). As with most treatments, success is most likely with correct patient selection, which is likely to be one of the benefits of OTC for a subset of those needing to consider fertility preservation for medical reasons.

Line 239 – X years -  need to insert number of years

Lines 272-273: Age is related to ovarian reserve [19], but individual variation is present, WHICH IS why ovarian reserve markers at the time of OTC could possibly help us to predict the efficacy of OTT in terms of pregnancy and longevity of the grafted tissue.

Author Response

Response to reviewer 2 comments:

Thank you for the review.

  1. Given the assumed target audience, I feel care must be taken to ensure the correct context for the presented data.  The paper starts with a strong statement that OTC is an established treatment (lines 20 and 41), and indeed the ASRM has stated this (ref 2 quotes ref 1). However, it is described as innovative treatment by ESHRE (Hum Reprod Open 2020 Nov 14;2020(4):hoaa052. doi: 10.1093/hropen/hoaa052.eCollection 2020.). In the UK, for example, funding is not routinely available for OTC (whereas it is for cryopreservation of eggs/embryos). NICE did not update their recommendation for cryopreservation to include OTC in their most recent update, although this was in 2017. Given that this paper is aimed at non-reproductive specialists, it is important that they are not led to believe that this is standard treatment for fertility preservation in most countries. Cryopreservation of eggs or embryos remains the standard (ASRM, ESHRE, NICE). The authors have chosen not to use such a strong statement in the simple summary.

Response to 1. Thank you for your consideration. We have modified the text according to your recommendations. 

The audience are likely to benefit from a brief discussion of the pros and cons of OTC vs standard fertility preservation options (cryopreservation of eggs/embryos) for context. It is important to consider the potential harm of the OTC procedure to future endocrine and fertility outcomes (particularly since 1 ovary was removed, halving the ovarian reserve). As with most treatments, success is most likely with correct patient selection, which is likely to be one of the benefits of OTC for a subset of those needing to consider fertility preservation for medical reasons.

Response to 2. I agree that the success and efficacy of OTC and OTT need to be evaluated in light of the total utilization rates of ovarian tissue and compared to the results of standard treatment with vitrification of oocytes and embryos. We have added a small section about this in the discussion. 

  1. Line 239 – X years -  need to insert number of years

Lines 272-273: Age is related to ovarian reserve [19], but individual variation is present, WHICH IS why ovarian reserve markers at the time of OTC could possibly help us to predict the efficacy of OTT in terms of pregnancy and longevity of the grafted tissue.

Response to 3. Thank you for the linguistic corrections. They are applied to the text.